# Finetuning Large Language Model as an Effective Symbolic Regressor

## Abstract

Deriving governing equations from observational data, known as Symbolic Regression (SR), is a cornerstone of scientific discovery. Large Language Models (LLMs) have shown promise in this task by leveraging their vast cross-disciplinary scientific knowledge. However, existing LLM-based methods primarily rely on direct inference or prompt engineering, often requiring excessive inference iterations to converge on correct formulas or failing to treating complex equation targets. These limitations in effectiveness and generalization stem from an inherent tension between pre-trained LLMs' proficiency in approximate reasoning and the high-precision demands of SR tasks. To bridge this gap, we propose to fine-tune LLMs for enhanced SR capability. Yet, the absence of dedicated datasets for SR-oriented fine-tuning remains a critical barrier. We thus introduce SymbArena, specifically engineered to optimize LLMs for SR. This benchmark comprises over 148,000 diverse equations formulated as corpora of 1.83 billion tokens for LLM utilization, enabling effective training and inference. Further, to ensure a more comprehensive and fair evaluation, SymbArena proposes a heuristics metric to precisely quantify form-level consistency, going beyond existing SR numerical-oriented evaluation strategies. With this benchmark, we explore mainstream LLM fine-tuning techniques for SR tasks and establish Symbolic-R1, a simple yet effective LLM-based SR strong baseline. Experimental results validate Symbolic-R1 as the first LLM to exceed traditional numerical methods in both numerical precision and symbolic form accuracy, outperforming the second-best LLM baseline with improvements of 2-fold gains in $R^2$ score and 10.3% in form-level consistency score. Code is available at https://anonymous.4open.science.

## 1 Introduction

Symbolic Regression (SR), which aims to derive the underlying governing equations from observational data, is a fundamental task in scientific discovery (Wang et al., 2023; Cornelio et al., 2023). The main challenge of the SR task is the difficulty of optimization for the regressed symbolic equation. Early solutions are usually based on numeral optimization, such as genetic programming (Schmidt & Lipson, 2009; Mei et al., 2022), discovering equation heuristically through evolutionary algorithms. With the development of deep learning, task-specific supervised learning methods achieve satisfactory progress in both effectiveness and accuracy (Biggio et al., 2021; Kamienny et al., 2022; Shojaee et al., 2023). Yet, obviously, a good symbolic regressor requires complex background knowledge to support equation discovery. Hence, in the LLM period, LLM-based SR has attracted more and more attentions.

Current LLM-based SR approaches, primarily leveraging direct inference or prompt engineering, operate by iteratively generating a set of candidate equations and retaining only the best-fitting one. Although the promising of the LLM having rich scentific prior, the LLMs often suffer from limitations in effectiveness and generalization, as shown in Fig. 1. The main reason for this phenomenon comes from an inherent tension between pre-trained LLMs' proficiency in approximate reasoning and the high-precision demands of SR tasks. Prevailing general-purpose Large Language Models (LLMs) are trained to generating ambiguous but diverse outputs to fulfilled human perception. However, the ambiguous and inaccuracy in the answer induces catastrophic precision degradation in the SR, as minor symbolic deviations (e.g., operator substitution $\times \rightarrow +$ or coefficient alteration $2.0 \rightarrow 2.1$) propagate into error accumulation, leading to physical error of derived equations. To bridge this gap,

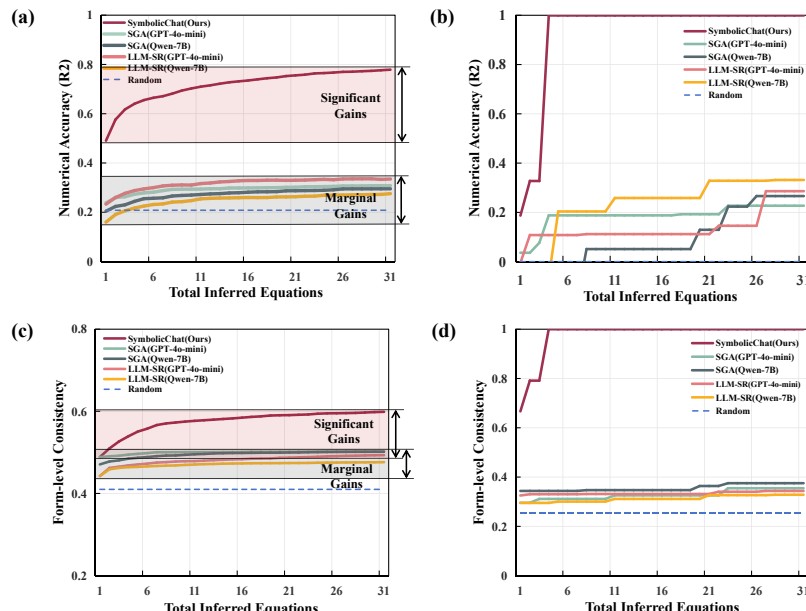

Figure 1: Comparison of our method against baselines on numerical accuracy $R^2$ and form-level consistency, showing both average performance (a, c) and a representative case study (b, d). The results reveal that the iterative approach of baseline methods is largely ineffective, with performance plateauing just above a random baseline. This suggests their process is closer to an exhaustive search than true inference. In contrast, our model achieves significant results in its first inference and reasons out the correct equation much faster (b, d). The substantial gains shown in (a) and (c) further confirm our model's high effectiveness and generalization.

fine-tuning is a optimal strategy which could introduce task-specific constraints and accurate knowledge (Hu et al., 2022; Bai et al., 2022; Rafailov et al., 2023; Han et al., 2024). Yet, the absence of dedicated datasets for SR-oriented fine-tuning remains a barrier.

To address this challenge, in this paper, we develop a new symbolic regression dataset and benchmark, termed SymbArena, to facilitate LLM-based SR fine-tuning. It comprises over 148,000 diverse equations, which collectively form a massive corpus of 1.83 billion tokens. The entire dataset covers a broad spectrum of mathematical structures and complexity levels and each equation is accompanied by a task instruction, its corresponding numerical data.To ensure a fair evaluation free from pre-training data contamination, we first confirm the novelty of our synthetically generated equations and then partition the dataset according to equation skeletons, preventing potential information leakage on the equation structural level across the training and test sets. Except that, for sufficient evaluation, the SymbArena assesses symbolic regression models on two crucial perspectives. On the one hand, the SymbArena introduces numerical-level evaluation metrics, error term $R^2$ and tolerance-based accuracy ($Acc_\tau$), following existing SR benchmarks to quantify data fitting fidelity, essentially the accuracy of the dependent variable output by the regressed formula. On the other hand, the SymbArena proposes to evaluate form-level consistency, including a LLM-based metric and a well-designed heuristic metric. The former utilizes LLM to measure the form similarity between the model output and reference, while the latter are LLM-independent, measuring substring similarity between predicted and ground-truth mathematical expression strings in their coefficient-abstracted canonical forms. This novel metric compensates for the limitations of conventional numerical metrics, where erroneous formulations artificially depress fitting errors through over-optimized coefficients, thereby masking the true discrepancies.

Building upon this dataset, we delve into mainstream LLM fine-tuning and inference techniques on the SR tasks and conclude as a Symbolic-R1, a strong LLM-based baseline specifically tailored for symbolic regression tasks. Symbolic-R1 leverages a novel Form-GRPO to operate the reinforcement fine-tuning (RFT) based on a series of manually designed form reward rules to guide

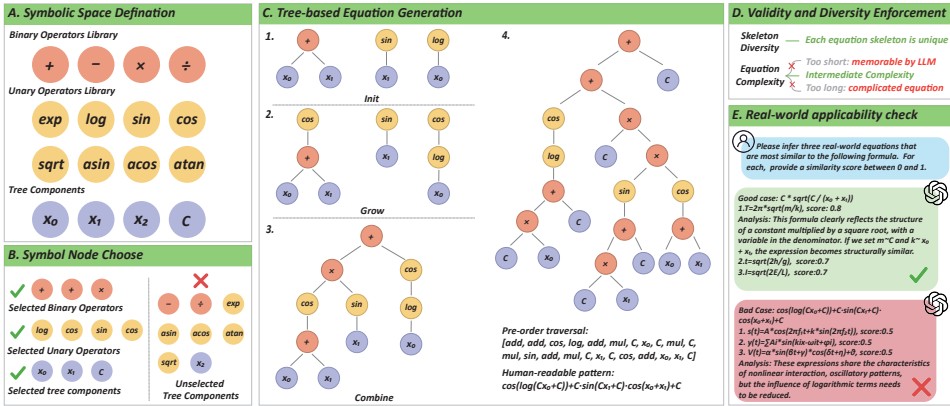

Figure 2: Workflow of equation generation. (A) Define the symbolic space with a library of operators and terminals. (B) Apply structural constraints to select valid components. (C) Construct tree-based expressions incrementally. (D) Enforce validity and filter by complexity. (E) Check real-world applicability via similarity scoring.

structure-aware generation and improves symbolic fidelity. Subsequently, we introduce a Hypothesis–Experiment–Revision (HER) inference framework, an interative strategy designed to yield more reliable equations. By circumventing the limitations of traditional model-based reward schemes, Symbolic-R1 achieves superior effectiveness and generalization, as shown in Fig. 1. Experimental results validate our Symbolic-R1 as the first LLM to exceed traditional numerical methods in both tolerance-based accuracy and form-level consistency, Furthermore, Symbolic-R1 significantly outperforms the next-best LLM baseline, with 2-fold improvements of $R^2$ score and 10.3% in form-level consistency score.

The main contributions of this paper are as follows:

- We introduce SymbArena, a large-scale symbolic regression benchmark with 148,102 diverse equations, featuring separate training and test splits.

- We design a new evaluation scheme that jointly considers form-level consistency and data fidelity. This design also helps to enables a more fine-grained analysis of model performance.

- We propose Symbolic-R1, a strong LLM-based baseline to handle symbolic regression tasks, which utilizes a novel Form-GRPO with a Hypothe- sis–Experiment–Revision inference framework to achieve a new state of the art.

## 2 SYMBARENA

This section details the construction of our proposed benchmark, SymbArena, encompassing equation generation, data generation and metric.

### 2.1 DATA GENERATION

The entire dataset $D$ contains several equations and corresponding data points, written as $D = \{(d_i, f_i)\}_{i=1}^{N}$, where $f_i$ is the equation form. $d_i \in \mathcal{R}^{K \times (C_i+1)}$ is the data matrix containing $K$ groups of data pair, each of which consists of $C_i$ independent variable values and one dependent variable value. To formulate such $D$, we follow two steps: 1) Generating equation $f_i$ and 2). Sample data points as $d_i$.

*Equation Generation.* The entire equation generation process is shown in Fig. 2. To generate equations in a program way, we represent each equation $f_i$ as a tree structure (Lample & Charton, 2019; Kamienny et al., 2022), where each tree node represents either a mathematical operator or a terminal symbol. Then, the equation is generated following:

- **Symbol Node Choose**: The equation is represented as a tree, including two kinds of nodes. One is the variable nodes and the other is the operation node. For the *variable* node, we sample a scalar as the number of the independent variable, $C_i$, formulating $C_i$ independent variable placeholder node. Also, we introduce a constant placeholder node to cover potential constants in the equation. For the *operation* node, we pre-define a library of mathematical symbols serving as basic node candidates, consisting of unary operators (e.g., `sin`, `log`) and binary operators (e.g., `+`, `*`). The unary terms focus on a single variable while the binary ones treat multiple variable relations. we randomly select multiple unary and binary operators to cover the potential concerned calculations.

- **Tree-based Equation Generation**: Getting all variable and operation nodes, we construct equations as trees through a recursive, incremental process: starting with a partial tree, we iteratively sample operators or terminals from symbol space and insert them into designated expansion points. This generative procedure, by construction, ensures that all synthesized equations comply with the rules of real formulas and do not contain mathematically unacceptable operations, such as single-sided brackets or plus signs followed by minus signs.

- **Unique and Complexity Check:** For each synthesized equation, we check its uniqueness by measuring the skeleton string similarity to make sure each equation is as unique as possible. Also, we check equation complexity by scanning layer number of each tree, and filter out over-simple (lower than 4) or over-complex (larger than 12) equations, make sure data do not overload model learning but also are not too simplistic for real application.

Table 1: Comparison of symbolic regression benchmarks. SymbArena distinguishes itself through its massive scale, the inclusion of a train/test split, and its support for both traditional and LLM-based methods.

| Benchmarks | Numbers of Equations | Train & Test | Evaluation Metrics | Supported Methods |
| --- | --- | --- | --- | --- |
| Nguyen | 12 | ✗ | R2 | Traditional only |
| R rational | 3 | ✗ | R2 | Traditional only |
| SRbench | 252 | ✗ | R2, Sympy Accuracy | Traditional only |
| LLM-SRbench | 239 | ✗ | NMSE, Acc, Symbolic Accuracy | LLM-based only |
| SymbArena | 148,102 | ✔ | R2, Acc, Symbolic Similarity | Both |

*Data Matrix Calculation.* For the data matrix $d_i$, we first randomly sample $K$ groups of $C_i$ independent variables values from one of two distributions randomly: uniform $\mathcal{U}(-dom, dom)$ or Gaussian $\mathcal{N}(0, dom)$ where $dom$ is a domain parameter controlling independent variable value range. With the sampled independent variables, we substitute them in the equation and get the dependent outputs, formulating with the independent variables as the data matrix $d_i$. In our practice, $dom$ is set as 10, providing a fixed range of independent variables, leading to stable training. A potential concern might be that dom could limit model generalization by restricting inputs to normalized data. However, the data normalization is not a fundamental limitation, as real-world applications can incorporate a denormalization step for the independent variables into the regressed equation if needed.

*Data Statistics.* As shown in Tab. 1, SymbArena is compared with several widely adopted SR benchmarks, including Nguyen (Uy et al., 2011), R rational (Krawiec & Pawlak, 2013), SRbench (La Cava et al., 2021), and LLM-SRbench (Shojaee et al., 2025). SymbArena contains 148,102 equations, significantly exceeding the size of existing datasets. Unlike the other datasets, SymbArena is designed for both training and testing. Among these, 147,590 equations are used for training and 512 for testing, allowing a standardized evaluation pipeline. In addition, SymbArena supports both traditional SR methods and LLM-based approaches, providing a unified platform for evaluating diverse SR paradigms.

*Training and Test Data Processing.* The aforementioned steps provide us with a symbolic equation set. Now, we split the dataset into a train split and a test split. Also, for the test set, we perform several operations to enhance the evaluation fairness and quality.

- **Train-Test split:** We calculate the unique form number of the entire dataset. Based on the unique form, we split training and testing sets based on the form, preventing potential information leakage of the equation form, such as formulas with the same form but different coefficients appear in both the training and test sets, respectively.

- **Reality Enhancement for Test set:** For the test set, we operate a reality enhancement. For each equation, we retrieve its 3 most similar human-known scientific equations based on the LLM deep

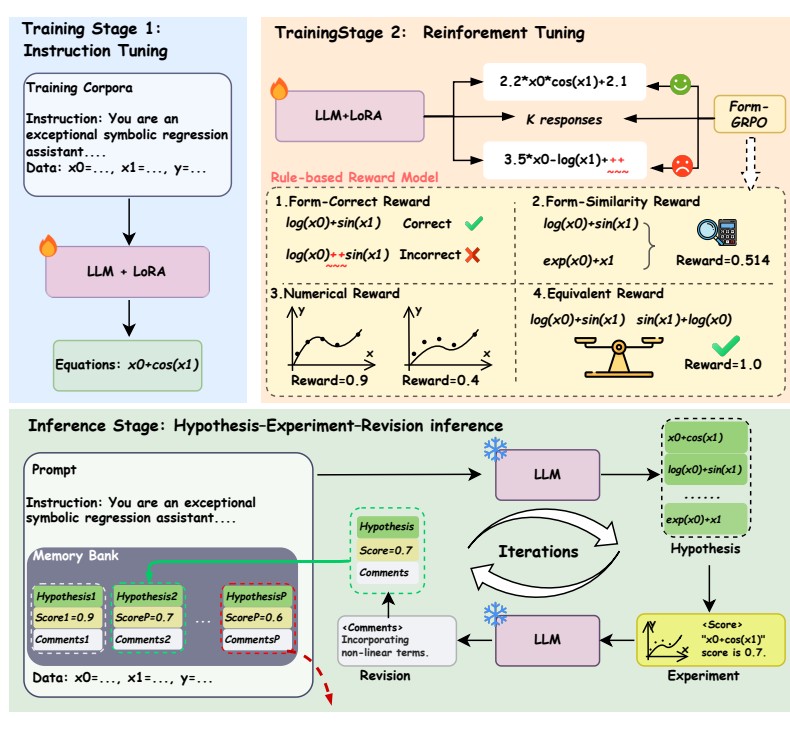

Figure 3: Overview of the proposed Symbolic-R1 framework. It consists of a two-stage training phase (1. Instruction Tuning; 2. Reinforcement Tuning with Form-GRPO) followed by an inference phase, where a Hypothesis–Experiment–Revision loop is used to refine equations.

research function. Also, we prompt the LLM to provide similarity between the retrieval query and the 3 return results with corresponding inference explanation. Based on that, we manually check the LLM retrieval with inference thoughts and filter out equations with low similarity with existing knowledge. This step is quite important to make sure the evaluation sample is close to real applications, rather than trivial simulation formulas out of touch with real scientific scenarios. Another point worth noting is that this strategy is also fairer than directly using real data for evaluation, because the formulas we retain do not exactly exist before, avoiding the possibility that LLM has seen them during the pre-training, more suitable for LLM evaluation.

## 2.2 METRIC

We evaluate the performance on the Symbolic Regression task from two primary perspectives: tolerance-based accuracy and form-level consistency.

1. **Numerical Accuracy**: Measures how closely the predicted outputs match the ground-truth values. Following the evaluation protocols proposed in prior studies (Kamienny et al., 2022; Biggio et al., 2021; La Cava et al., 2021), we adopt two widely used metrics to assess numerical accuracy: the coefficient of determination ($R^2$) and tolerance-based accuracy ($\text{Acc}_\tau$), which reports whether the worst-case relative error is within tolerance $\tau$, given by:

$$R^2 = 1 - \frac{\sum_{i=1}^{N_{\text{test}}} \left( f(x_i) - \hat{f}(x_i) \right)^2}{\sum_{i=1}^{N_{\text{test}}} \left( f(x_i) - \overline{f(x_i)} \right)^2}, \tag{1}$$

$$\text{Acc}_\tau = \mathbb{1}\left( \max_{1 \leq i \leq N_{\text{test}}} \left| \frac{\hat{f}(x_i) - f(x_i)}{f(x_i)} \right| \leq \tau \right), \tag{2}$$

where $\hat{f}(x_i)$ and $f(x_i)$ are the dependent outputs generated by feeding the same data into the predicted and ground-truth equations.

2. **Form-level Consistency**: Evaluates whether the skeleton structure of the predicted equation is consistent with the ground truth. To achieve this, we propose a comprehensive, multi-faceted Form-level Consistency metric. Specifically, we first extract the formula skeletons from the original equations. We then quantify the similarity by first decomposing each skeleton into a vector of six key structural features: operators, functions, variables, constants, structural pattern, and complexity. The similarity for each feature pair is calculated individually using methods like the Jaccard index for sets and relative ratios for counts. The final form-level consistency score, $S_{\text{struct}}$, is then computed as the average of these individual component similarities:

$$S_{\text{struct}} = \frac{1}{N_{\text{test}}} \sum_{i=1}^{N_{\text{test}}} \text{Sim}(\hat{f}i, fi), \tag{3}$$

The details of the similarity function can be seen in the Supplementary. where $\hat{f}_i$ and $f_i$ are the predicted and ground-truth equations, respectively. Moreover, we leverage GPT-4o as a semantic adjudicator (Achiam et al., 2023) to serve as a scalable, cost-effective proxy for human expert evaluation, providing a holistic consistency score from 0 to 1 based on the structural correspondence between the predicted and ground-truth skeletons. By introducing these two flexible scoring mechanism, we address a key limitation of previous work (La Cava et al., 2021; Shojaee et al., 2025), which often relies on a binary notion of correctness (i.e., equivalent or not) and thus fails to capture the fine-grained form-level consistency between equations. Such graded consistency measures can offer deeper insights into where symbolic regression models succeed or fail.

## 3 METHOD

In this section, we propose a strong LLM-based SR baseline, Symbolic-R1. Its main idea is to fine-tune the SR data for the LLM to drive highly effective iterative inference. The entire pipeline is shown in Figure 3. The model is first fine-tuned by instruction tuning (Sec. 4.1), followed by reinforcement fine-tuning with Form-GRPO (Sec. 4.2). During inference, the model utilizes the iterative reflection framework (Sec. 4.3) to refine outputs, leading to accurate equations.

### 3.1 INSTRUCTION TUNING

For a SR dataset $D = \{(d_i, f_i)\}_{i=1}^{N_{\text{train}}}$, we first formulate the data matrix $d_i$ as a input representation $P_i$ involves two components: an data-shared instruction part $I$ and a data-specific value part $V_i$. The instruction $I$ consists of: **1).** The definition of the SR task that requests the LLM to derive a symbolic equation from the provided data and a corresponding example, e.g., data to 'f(x) = 2.0*sin(x) + 3.0'. **2).** Basic operation scheme introduction, like existing commonly used symbolic operations, e.g., the definition of '+' and '-'. The value part $V_i$ is formed by reformatting the $d_i$ as input-output pairs $\{(x_i, f(x_i))\}_{i=1}^{K}$ into a key-value structure (e.g., x_0=..., x_1=..., f(x)=...) where $x_i \in \mathcal{R}^{K \times C_i}$ is the independent variable values and $f(x_i) \in \mathcal{R}^K$ is the dependent one.

The input $P_i$ is fed into the LLM backbone, which in turn generates the predicted equation $\hat{f}_i$. Similar to most language models, we employ cross-entropy loss which constrains the similarity between estimated and ground-truth tokens, which can be presented as:

$$\mathcal{L} = \mathbb{E}_{i=1}^{N_{\text{train}}} \text{Cross\_Entropy}(\hat{f}_i, f_i). \tag{4}$$

For parameter-efficient fine-tuning, we employ Low-Rank Adaptation (LoRA) (Hu et al., 2022) on the LLM backbone.

### 3.2 REINFORCEMENT TUNING BY FORM-GRPO

After the instruction tuning stage, we devise a Form-GRPO scheme to further optimize the LLM. Rewards are specifically designed to guide the LLM towards generating expressions with enhanced structural similarity and numerical accuracy. Our reward system comprises four types of rewards: Form-Correct Reward, Form-Similarity Reward, Numerical Reward and Equivalent Reward.

**Form-Correct Reward.** We define the format reward $\mathcal{R}_{\text{format}}$ to penalize the equation syntactically conflicting with the mathematical rule. To achieve this, we establish a valid decision function, `is_valid`. For each equation, it tries to convert it into an equivalent operation from specified numerical libraries (e.g., NumPy or math) through dynamic code generation. If the transformation failed, we tag the generated equation with a negative reward as follows:

$$\mathcal{R}_{\text{format}}(\hat{f}_i) = \begin{cases} 1.0, & \texttt{is\_valid}(\hat{f}_i) \\ -1.0, & \text{otherwise} \end{cases}. \tag{5}$$

**Form similarity Reward.** To introduce the form-level consistency supervision, we directly utilize the aforementioned Form similarity described in Equation 3 as the Form similarity Reward.

**Numerical Reward.** We use the numerical reward $\mathcal{R}_{\text{numerical}}$ to quantify how well an equation fits the input-output data. This reward is only utilized for equations passed the check of the valid decision function. We use the $R^2$ score to achieve the numerical reward. However, since the raw $R^2$ score is unbounded from below, its direct use can lead to training instability. We therefore apply a truncation strategy, defining the final reward as follows:

$$\mathcal{R}_{\text{numerical}}(\hat{f}_i) = \begin{cases} \max(0, R^2(\hat{f}_i, f_i)), & \text{if } \texttt{is\_valid}(\hat{f}_i) \\ 0, & \text{otherwise} \end{cases}, \tag{6}$$

This ensures that only syntactically correct equations are eligible for a numerical reward, while invalid ones are penalized with a score of zero.

**Equivalent Reward.** We define the equivalence reward $\mathcal{R}_{\text{equiv}}$ to incentivize the generated equations that have a completely equivalent form to the ground truths (do not need the same coefficients). Given the difficulty and critical importance of achieving a perfect structural match, we apply a significant bonus for this accomplishment. The reward is computed as follows:

$$\mathcal{R}_{\text{equiv}}(\hat{f}_i, f_i)) = \begin{cases} 1.0, & \text{if } \hat{e}_i = e_i \\ 0.0, & \text{otherwise} \end{cases}, \tag{7}$$

where $\hat{e}_i$ and $e_i$ represent the skeletons extracted by replacing the coefficients in the predicted equation $\hat{f}_i$ and the ground-truth equation $e_i$ as placeholders, respectively.

The final reward function, $\mathcal{R}$, is defined as a weighted sum of these components:

$$\mathcal{R} = w_1 \mathcal{R}_{\text{format}} + w_2 \mathcal{R}_{\text{similarity}} + w_3 \mathcal{R}_{\text{numerical}} + w_4 \mathcal{R}_{\text{equiv}}. \tag{8}$$

To leverage this composite reward signal for policy optimization, we employ the Group Relative Policy Optimization (GRPO) algorithm (Shao et al., 2024). Specifically, for a given input, the process begins by sampling a group of $G$ candidate equations from the reference policy, where $G$ is set as 8 following the GRPO original setting. Each equation is then evaluated to obtain rewards, which are subsequently normalized using the group's mean and standard deviation. This single normalized reward serves as the advantage estimate for all tokens within the corresponding equation.

### 3.3 Hypothesis–Experiment–Revision inference

During inference, we introduce a Hypothesis–Experiment–Revision (HER) framework, an iterative strategy designed to yield more reliable equations. The core idea is to generate multiple candidate hypotheses in each iteration, validate them through quantitative experiments, and incorporate reflective revision to selectively retain high-quality candidates—thereby emulating the scientific cycle of hypothesis formulation, experimental validation, and reflective refinement.

**Hypothesis:** Given an input $P$, the Symbolic-R1 model produces a set of candidate equations $\{\hat{f}_1, \hat{f}_2, \ldots, \hat{f}_K\}$, where $K$ is the number of hypotheses (set to 6, following prior multi-hypothesis discovery approaches such as SGA (Ma et al., 2024)). For each candidate, we employ numerical optimization tools to refine the coefficients and improve predictive accuracy.

**Experiment:** Each hypothesis is then quantitatively evaluated using the $R^2$ score, which serves as an analogue to experimental validation of the consistency between a hypothesis and real-world observations.

Table 2: Evaluation results on SymbArena. The **best** and second-best figures are in bold and underlined, respectively.

| Methods | Type | $S_{\text{struct}}$ (gpt-4o) | $S_{\text{struct}}$ (rule) | $R^2$ | $\text{Acc}_\tau$ |
|---|---|---|---|---|---|
| gplearn | GP | 0.266 | 0.296 | 0.200 | 0.074 |
| AFP | GP | 0.264 | 0.371 | 0.245 | 0.060 |
| AFP-FE | GP | 0.298 | 0.389 | 0.296 | 0.072 |
| GP-GOMEA | GP | 0.337 | 0.356 | 0.360 | 0.238 |
| uDSR | GP | 0.244 | 0.422 | 0.349 | 0.027 |
| PySR | GP | 0.368 | 0.382 | 0.663 | 0.398 |
| LLM-SR(gpt-3.5-turbo) | LLM-based | 0.291 | 0.437 | 0.255 | 0.189 |
| LLM-SR(gpt-4o-mini) | LLM-based | 0.299 | 0.415 | 0.355 | 0.221 |
| LLM-SR(Qwen2.5-7B) | LLM-based | 0.306 | 0.404 | 0.313 | 0.205 |
| SGA(gpt-3.5-turbo) | LLM-based | 0.342 | 0.504 | 0.327 | 0.170 |
| SGA(gpt-4o-mini) | LLM-based | 0.351 | 0.492 | 0.286 | 0.168 |
| SGA(Qwen2.5-7B) | LLM-based | 0.362 | 0.480 | 0.273 | 0.176 |
| **Symbolic-R1** | LLM-based | **0.436** | **0.607** | **0.808** | **0.404** |

Table 3: Ablation Study of Symbolic-R1. This table evaluates the individual contribution of each core component: instruction tuning (IFT), reinforcement tuning (RFT), and the Hypothesis–Experiment–Revision (HER) framework. The **best** and second-best figures are in bold and underlined, respectively.

| LLM Backbone | Iteration Strategy | $S_{\text{struct}}$ (gpt-4o) | $S_{\text{struct}}$ (rule) | $R^2$ | $\text{Acc}_\tau$ |
|---|---|---|---|---|---|
| Qwen2.5-7B | None | 0.353 | 0.466 | 0.201 | 0.145 |
| | LLM-SR | 0.306 | 0.404 | 0.313 | 0.205 |
| | SGA | 0.362 | 0.480 | 0.273 | 0.176 |
| | HER | 0.323 | 0.518 | 0.344 | 0.207 |
| +IFT | None | 0.406 | 0.590 | 0.313 | 0.193 |
| | LLM-SR | 0.439 | 0.601 | 0.618 | 0.310 |
| | SGA | 0.452 | 0.601 | 0.605 | 0.314 |
| | HER | **0.456** | **0.613** | 0.624 | 0.309 |
| +IFT +RFT | None | 0.403 | 0.574 | 0.540 | 0.252 |
| | LLM-SR | 0.396 | 0.538 | 0.743 | 0.343 |
| | SGA | 0.410 | 0.569 | 0.778 | 0.373 |
| | HER | 0.436 | 0.607 | **0.808** | **0.404** |

**Revision:** The experimental outcomes are subsequently fed back into Symbolic-R1, prompting it to generate qualitative assessments that resemble scientists' reflective notes on each hypothesis. These evaluations are jointly stored as tuples $(equation, score, comments)$ in a memory bank. The bank maintains the top-5 records ranked by $R^2$, updating in each iteration by discarding weaker candidates and preserving the most promising ones.

The curated memory is then reformulated into a reflection prompt and appended to the input $P$, enabling Symbolic-R1 to generate refined hypotheses in the next iteration. This iterative loop gradually improves the reliability and scientific plausibility of the inferred equations.

## 4 EXPERIMENTS

### 4.1 IMPLEMENTATION DETAILS

We employ Qwen2.5-7B-Instruct (Team, 2024) as our LLM backbone. For constructing all prompts and evaluating baseline methods, we consistently sample 200 input-output $(x, y)$ pairs from the dataset. During the instruction tuning stage, we utilize a dataset of 146,590 unique expressions, creating five distinct prompt variations for each, and fine-tune the model using LoRA with a rank of 8 and an alpha of 32. For the reinforcement tuning stage, we also use a set of 1,000 expressions and sample 8 candidate responses from the LLM for each input. We configure the reward function

hyperparameters to 1.0, 2.0, 2.0, and 4.0, with justification provided in the Appendix. All other settings are kept identical to the preceding stage. Finally, in the Hypothesis–Experiment–Revision inference stage, we configure the llm to run for 5 iterations, generating 6 equations within each cycle. More implementation and training details are provided in the supplementary material.

### 4.2 COMPARISON WITH STATE-OF-THE-ART METHODS.

As detailed in Table 2, our proposed method, Symbolic-R1, outperforms existing baselines across most key evaluation metrics. Existing LLM-based SR methods often failed to compare with traditional methods, proving the barrier of treating complex equations introduced by the gap between the pre-trained smooth-oriented knowledge and the SR's high accuracy demands. Fortunately, our Symbolic-R1 suppresses the traditional state-of-the-art PySR Cranmer (2023) on structure metrics and $\mathcal{R}^2$ by a large margin, also achieving comparable results on Acc$_\tau$, proving the effectiveness of Symbolic-R1 Compared with the previous state-of-the-art LLM-based SR method, LLM-SR Shojaee et al. (2024), our model achieves at least 0.1 structure increases and brings about 2-fold $R^2$ gains, only with one-fourth of the inference time cost. Also, compared with a state-of-the-art LLM science discovery model, SGA, our methods provide more advantages, further demonstrating the crucial importance of the fine-tuning for enhancing the symbolic regression ability of LLM in SR tasks.

### 4.3 ABLATION STUDY

To systematically evaluate the contributions of each core component in the Symbolic-R1 framework, we conducted a series of detailed ablation studies. These studies were designed to quantify the independent effectiveness of instruction tuning, reinforcement tuning, and the Hypothesis–Experiment–Revision framework. Our analysis begins with the untuned Qwen2.5-7B backbone, which serves as a baseline by achieving an $R^2$ score of 0.201. This initial result indicates that a general-purpose LLM, despite its foundational capabilities, is insufficient for high-precision symbolic regression. Furthermore, directly applying iterative strategies like SGA to this base model did not yield effective performance improvements.

With the introduction of instruction tuning, the model showed significant improvement, with its $R^2$ score increasing to 0.313 and its rule-based form-level consistency score rising substantially from 0.466 to 0.590. This demonstrates the critical role of this step in adapting the model to the task's format and enabling it to generate structurally correct expressions. Building on this foundation, we introduced reinforcement tuning using the GRPO algorithm, which led to a substantial leap in performance, boosting the $R^2$ score from 0.201 to 0.540. This provides strong evidence that our designed reward mechanism effectively guides the model to generate expressions that are not only structurally sound but also numerically fit the given data with high fidelity. Finally, after applying the Hypothesis–Experiment–Revision framework, the complete Symbolic-R1 model reached its peak performance, with the $R^2$ score further increasing from 0.540 to 0.808. This shows that using the fine-tuned LLM as a core generator and meticulously refining solutions through an iterative search of the solution space is the decisive step in achieving optimal numerical fitting and elevating the model's performance to a new state-of-the-art.

## 5 CONCLUSION

In this work, we delve into the limitations of the effectiveness and accuracy of LLM-based symbolic regression methods (see Appendix B for a full review) as the lack of SR-specific fine-tuning. To tackle this, we propose a SymbArena specifically designed for LLM fine-tuning, associated with a novel evaluation metric beyond traditional numerical and covering form-level consistency. With this, we propose a new LLM-based SR strong baseline, Symbolic-R1, achieving a new state of the art of LLM-based SR and suppressing traditional SR methods. Through these efforts, we hope to make modest contributions to the field of LLM scientific discovery, fostering the development of more robust and adaptable tools for future symbolic regression.

ETHICS STATEMENT

The authors acknowledge their responsibility to adhere to the ICLR Code of Ethics.

REPRODUCIBILITY STATEMENT

To ensure the reproducibility of our results, we provide comprehensive details of our methodology, experiments, and implementation.

- **Code and Data Availability:** The source code for this paper has been made publicly available at https://anonymous.4open.science. The corresponding datasets will be released upon acceptance to ensure full reproducibility.
- **Implementation Details:** A full description of our model architectures, algorithms, and experimental setup is provided in Appendix.

We believe this provides sufficient information for the research community to reproduce and build upon our findings.

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

# APPENDIX

## A  THE USE OF LARGE LANGUAGE MODELS (LLMs)

We acknowledge the use of a large language model (LLM) to improving grammar and wording of our paper. The authors are fully responsible for the content of this work.

## B  RELATED WORK

### B.1  TRADITIONAL SYMBOLIC REGRESSION

Symbolic regression is a machine learning technique for discovering concise, interpretable mathematical equations from data (Wang et al., 2019). Early approaches were dominated by Genetic Programming (GP), which proved effective in identifying physical laws from experimental data (Schmidt & Lipson, 2009). To overcome the efficiency and scalability limitations of GP, methods based on sparse regression were developed. A prominent example is the Sparse Identification of Nonlinear Dynamics (SINDy) algorithm, which identifies governing equations by applying sparse optimization to a library of candidate functions (Brunton et al., 2016; Rudy et al., 2017). More recently, deep learning has catalyzed major advances across the entire SR workflow. For instance, researchers use reinforcement learning to frame equation generation as a sequential decision problem (Li et al., 2025b; Xu et al., 2024) or apply pre-trained transformers to rapidly produce candidate equation structures (Biggio et al., 2021; Kamienny et al., 2022; Shojaee et al., 2023; Valipour et al., 2021; Vastl et al., 2024). Deep learning also enhances foundational aspects; autoencoders can learn low-dimensional latent data representations, facilitating the discovery of simpler models in an optimal coordinate system (Champion et al., 2019; Li et al., 2025a). Furthermore, hybrid methods show great promise, such as using RNNs to generate high-quality initial populations for more efficient genetic programming searches (Mundhenk et al., 2021; Landajuela et al., 2022).

### B.2  LLM FOR SYMBOLIC REGRESSION

Recent SR methods leverage LLMs as hypothesis generators, iteratively refining equations with data-driven feedback to embed physical knowledge (Li et al., 2025a). LLM-SR (Shojaee et al., 2024) uses the LLM as a black-box optimizer for self-improvement or as an evolutionary engine performing crossover and mutation. SGA (Ma et al., 2024) employs a bilevel optimization where the LLM generates discrete equation structures (upper level) and a differentiable simulator optimizes their continuous parameters (lower level). In contrast, LaSR (Grayeli et al., 2024) uses an LLM to evolve a library of abstract textual concepts that guide and accelerate a separate evolutionary search. However, a recent benchmark (Shojaee et al., 2025) reveals a key limitation: the performance of general-purpose LLMs plummets when they are prevented from reciting known equations. This highlights the need for domain-specific model adaptation over relying on pure inference.

## C  IMPLEMENTATION DETAILS

### C.1  EXPERIMENTAL ENVIRONMENT SETUP

We ensure reproducibility by providing the experimental environment and computational resources. Tab. 4 shows the environment configuration.

Table 4: Experimental Environment Setup.

| Component | Version |
| --- | --- |
| OS | Ubuntu 20.04.6 LTS |
| Python | 3.11.8 |
| PyTorch | 2.7.0 |
| Cuda | 12.4.1 |

## C.2 TRADITIONAL SR METHODS

We compare our Symbolic-R1 against several state-of-the-art traditional Symbolic Regression (SR) baselines, including gplearn, AFP (Schmidt & Lipson, 2010), AFP-FE (Schmidt & Lipson, 2009), GP-GOMEA (Virgolin et al., 2021), uDSR (Landajuela et al., 2022), and PySR (Cranmer, 2023). The hyperparameters for all baseline methods are set to the default values as specified in their original publications. A detailed summary of these parameters is provided in Tab. 5:

Table 5: Hyper-parameter setting of traditional SR methods.

| Methods | Hyper-parameters |
|---------|------------------|
| gplearn | {population_size: 1000, generations: 20, p_crossover: 0.9, max_samples: 1.0, parsimony_coefficient: 0.001} |
| AFP | {num_islands: 10, island_gens: 100, max_len: 64, max_len_init: 20, time_limit: 7200, popsize: 100, g: 50} |
| AFP-FE | {num_islands: 10, island_gens: 100, max_len: 64, max_len_init: 20, time_limit: 7200, popsize: 100, g: 50, FE_pop_size: 100, FE_ind_size: 10, FE_train_size: 10, FE_train_gens: 10} |
| GP-GOMEA | {generations: -1, initmaxtreeheight: 3, popsize: 32} |
| uDSR | {length_min: 4, length_max: 64, repeat_max: 3, soft_length_loc:10 , soft_length_scale: 5 } |
| PySR | {maxsize: 20, niterations: 40} |

## C.3 LLM-BASED SR METHODS

We implement two state-of-the-art LLM-based baselines:LLM-SR and SGA, each tested on SymbArena with three different LLM backbones: an open-source model (Qwen2.5-7B-Instruct) and two closed-source models (GPT-3.5-turbo and GPT-4o-mini). For LLM-SR, the maximum number of sampled equations is capped at 50. For SGA, the process is configured for 5 iterations, with each iteration involving the exploitation of two equations and the exploration of four. Across all experiments, the temperature for the LLM backbones is set to 0.7.

## C.4 PROMPTS

To ensure consistency, the prompts for all LLM-based methods follow the message-based prompting format as defined in the OpenAI Chat Completion API. A concrete example of the prompt template used is provided below.

### C.4.1 SYSTEM PROMPT:

```
You are an exceptional symbolic regression assistant.
Your specialty lies in analyzing numerical relationships among data and
variables.
When provided with mathematical questions or data from humans, you
carefully comprehend the essence of the problem, methodically clarify
relationships among variables.
Ultimately, you output a precise, concise, and interpretable
mathematical formula.
```

### C.4.2 MESSAGE

**1) Instruction Prompt:**

```
You will be provided with a set of input-output pairs.
Based on these data, infer the mathematical relationship between y and
multiple input variables.
Please note that the possible mathematical operations include:  +, -,
*, /, exp, log, sqrt, sin, arcsin, and constant terms.
```

**2) Memory Prompt (using in Symbolic-R1):**

```
You can refer to the previously proposed formulas and their
corresponding fitness scores (lower is better), which are stored in
pred_dict:
```

$$0 : [3\sin(x_0) + x_1 + 2.4, \quad 0.01]$$

```
    ...

Based on the analysis, here are some suggestions for improvement:
```

- Consider adding a term that captures the interaction between $x_0$ and $x_1$, such as $0.479 - 0.476x_1 - 0.231x_0$, to refine the model.

- Introduce a quadratic term for $x_0$ to capture non-linear effects, for example, $0.21 - 0.77x_0^2$.

```
Please consider these suggestions when generating new formulas.
```

**3) Data Prompt:**

```
The input sample data are as follows:
```
$$x_1 =, \quad x_2 =, \quad y =$$
$$x_1 =, \quad x_2 =, \quad y =$$
$$x_1 =, \quad x_2 =, \quad y =$$
```
    ...
Based on the above data, please infer the possible formula.
Ensure that your inference applies to all the provided data points, and
consider both linear and nonlinear combinations.
Verify whether your formula applies to the following new data point and
adjust it to ensure accuracy:
```
$$x_1 =, \quad x_2 =, \quad y =$$
```
    ...
Finally, please output only the formula string you inferred (e.g.
```
$y = 2.52 * x_0 + x_1 + 5.4$`), without any additional information.
```
Do not include any explanation, text, or extra information, only return
the expression string.
```

## C.5   SIMILARITY METRIC

Here, we provide a detailed breakdown of the calculation process for our rule-based form-level consistency metric, $S_{\text{struct}}(rule)$. This metric decomposes each equation into a vector of six key structural features and calculates a final score based on their aggregated similarity. The process is detailed below.

**1. Feature Extraction**   For both the predicted and ground-truth equations, we first extract a feature vector comprising six components:

- **Operators**: The set of unique mathematical operators used in the equation (e.g., $\{+, *, /\}$).

- **Functions**: The set of unique mathematical functions in the equation(e.g., $\{\sin, \arctan, \exp\}$).

- **Variables**: The set of unique variables in the equation(e.g., $\{x_0, x_1\}$).

- **Constants**: The count of the constant placeholder 'C'.

- **Structural Pattern**: A normalized representation of the equation's structure, generated by replacing all variables with a `VAR` placeholder. For example, the equation $C \cdot x_0^2 + C$ would yield the pattern $C \cdot \text{VAR}^2 + C$.

- **Complexity**: A score defined as the sum of three components: operators, functions and the maximum nesting depth of parentheses in the equation.

**2. Component-wise Similarity Calculation**   Next, we compute the similarity for each of the six feature pairs individually:

- **Operator, Function, and Variable Similarity**: For features represented as sets (Operators, Functions, Variables), we use the Jaccard Index to quantify their similarity. Given two sets $A$ and $B$, the Jaccard similarity is defined as:

$$\text{Sim}_{\text{Jaccard}}(A, B) = \frac{|A \cap B|}{|A \cup B|}, \tag{9}$$

- **Constant and Complexity Similarity**: For numerical features (Constant count and Complexity score), the similarity is calculated as the ratio of the minimum value to the maximum value. This ensures the score is bounded between 0 and 1. For two values $v_1$ and $v_2$:

$$\text{Sim}_{\text{ratio}}(v_1, v_2) = \frac{\min(v_1, v_2)}{\max(v_1, v_2)}, \tag{10}$$

If both values are zero, the similarity is defined as 1.

- **Structural Pattern Similarity**: The similarity between two structural pattern strings is calculated based on a character-wise comparison. It is defined as the number of matching characters at identical positions, normalized by the length of the longer pattern string.

**3. Final Score Aggregation**   The final Form-level Consistency score $S_{\text{struct}}$ is computed as the unweighted average of these six individual component similarities. This approach provides a balanced assessment of structural alignment without introducing subjective bias from manual weighting. The formula is:

$$Sim = \frac{1}{6} \sum_{k=1}^{6} \text{Sim}_k, \tag{11}$$

where $\text{Sim}_k$ represents the similarity score for the k-th structural feature. The final score is clipped to the range $[0, 1]$.

## D   MORE EXPERIMENTS AND VISUALIZATIONS

### D.1   ABLATION STUDY ON FORM-GRPO HYPERPARAMETER

To determine the optimal composition of our reward function, we conducted a series of experiments to analyze the impact of different reward hyperparameter weights on model performance. As detailed in Tab. 6, we evaluated six distinct configurations by varying the weights for form similarity reward(`R_similarity`), numerical reward (`R_numerical`), and equivalent reward (`R_equiv`). The form-correct reward weight (`R_format`) was held constant at 1.0 across all experiments, as we consider format correctness a foundational requirement.

Our analysis began with a baseline configuration (a), where all reward components were equally weighted, yielding an $R^2$ score of 0.751. While increasing only the numerical reward weight (configuration b) resulted in the highest $R^2$ (0.814) and $\text{Acc}_r$ (0.412), we observed that this came at the cost of structural quality, as indicated by lower $S_{\text{struct}}$ scores.

Our goal was to find a configuration that achieves a robust balance across all performance dimensions. Through systematic adjustments, we identified configuration (f), with weights of 1.0, 2.0, 2.0, and 4.0 for `R_format`, `R_similarity`, `R_numerical`, and `R_equiv` respectively. This configuration achieved the highest scores for structural similarity ($S_{\text{struct}}$(gpt-4o) = 0.436 and $S_{\text{struct}}$(rule) = 0.607) among all tested setups.

Table 6: Ablation study on Form-GRPO hyperparameter settings on model performance. Each row represents a different configuration of reward weights.

| Type | $R_{format}$ | $R_{similarity}$ | $R_{numerical}$ | $R_{equiv}$ | $S_{\text{struct}}$ (gpt-4o) | $S_{\text{struct}}$ (rule) | $R^2$ | $Acc_\tau$ |
|------|------|------|------|------|------|------|------|------|
| (a) | 1.0 | 1.0 | 1.0 | 1.0 | 0.431 | 0.598 | 0.751 | 0.379 |
| (b) | 1.0 | 1.0 | 2.0 | 1.0 | 0.405 | 0.580 | 0.814 | 0.412 |
| (c) | 1.0 | 2.0 | 1.0 | 1.0 | 0.435 | 0.606 | 0.738 | 0.375 |
| (d) | 1.0 | 2.0 | 2.0 | 1.0 | 0.435 | 0.605 | 0.794 | 0.391 |
| (e) | 1.0 | 2.0 | 2.0 | 2.0 | 0.433 | 0.607 | 0.797 | 0.392 |
| (f) | 1.0 | 2.0 | 2.0 | 4.0 | 0.436 | 0.607 | 0.808 | 0.404 |

Although configuration (f) exhibits a marginal decrease in $R^2$ (0.808) and $Acc_r$ (0.404) compared to the peak values in configuration (b), its superior performance in structural metrics demonstrates a more desirable trade-off. The strong emphasis on equivalent reward (`R_equiv` = 4.0) proved crucial for enhancing the model's ability to generate outputs that are not only numerically accurate but also logically and structurally sound.

Therefore, we concluded that configuration (f) represents the optimal balance for our objectives. These hyperparameter weights were adopted for all main experiments.

## D.2 EVALUATING THE IMPACT OF NOISE ON MODEL PERFORMANCE

To evaluate the model's robustness against measurement uncertainties, we introduce additive zero-mean Gaussian noise to the output values ($y_i$) of the test set. The standard deviation of the noise is fixed at $\sigma = 0.001$. This procedure is designed to simulate a low level of absolute measurement error and examine the model's stability under this specific condition.

Table 7: Evaluation of model robustness to additive Gaussian noise ($\sigma = 0.001$) on the SymbArena dataset. The **best** and second-best results are highlighted in bold and underlined, respectively.

| Methods | Type | $S_{\text{struct}}$ (gpt-4o) | $S_{\text{struct}}$ (rule) | $R^2$ | $Acc_\tau$ |
|------|------|------|------|------|------|
| gplearn | GP | 0.261 | 0.301 | 0.209 | 0.074 |
| AFP | GP | 0.293 | 0.396 | 0.276 | 0.076 |
| AFP-FE | GP | 0.297 | 0.394 | 0.246 | 0.058 |
| GP-GOMEA | GP | 0.337 | 0.362 | 0.334 | 0.214 |
| PySR | GP | 0.371 | 0.377 | 0.625 | 0.341 |
| LLM-SR(gpt-4o-mini) | LLM-based | 0.343 | 0.499 | 0.251 | 0.169 |
| LLM-SR(Qwen2.5-7B) | LLM-based | 0.340 | 0.452 | 0.250 | 0.159 |
| SGA(gpt-4o-mini) | LLM-based | 0.342 | 0.505 | 0.271 | 0.193 |
| SGA(Qwen2.5-7B) | LLM-based | 0.352 | 0.488 | 0.253 | 0.175 |
| **Symbolic-R1** | LLM-based | **0.403** | **0.596** | **0.793** | **0.353** |

As presented in Tab. 7, the introduction of noise led to a predictable performance degradation across most baseline methods. Other competitive methods such as PySR experienced a more pronounced performance drop (e.g., its $R^2$ score fell from 0.663 to 0.625). In contrast, Symbolic-R1 demonstrated exceptional stability. Specifically, its performance exhibited only a marginal decline: the $R^2$ score decreased minimally from 0.808 to 0.793, and the rule-based structural similarity, $S_{\text{struct}}$(rule), dropped from 0.607 to just 0.596. This minimal decay underscores the model's high resilience to low-level data perturbations.

## D.3 EVALUATION ON MORE DATASETS

To further assess the generalization capability of our proposed method, its performance was also evaluated on a set of five well-established benchmarks, including Nguyen (Uy et al., 2011), Constant (Tian et al., 2025), Keijzer (Keijzer, 2003), R rational (Krawiec & Pawlak, 2013), and SR-Bench (La Cava et al., 2021).

Table 8: Performance comparison on five well-established benchmarks. The **best** and second-best results are highlighted in bold and underlined, respectively.

| Method | Nguyen | Constant | R | Keijzer | SRBench | Overall avg. |
|---|---|---|---|---|---|---|
| gplearn | 0.767 | 0.606 | 0.354 | 0.638 | 0.551 | 0.621 |
| AFP | 0.702 | 0.720 | 0.657 | 0.800 | 0.755 | 0.742 |
| AFP-FE | 0.832 | 0.790 | 0.457 | 0.783 | 0.619 | 0.727 |
| GP-GOMEA | 0.597 | 0.675 | 0.336 | 0.261 | 0.684 | 0.539 |
| PySR | 0.950 | **0.998** | **0.997** | **0.999** | 0.937 | 0.968 |
| LLM-SR(gpt-4o-mini) | 0.567 | 0.620 | 0.051 | 0.523 | 0.234 | 0.434 |
| LLM-SR(Qwen2.5-7B) | 0.623 | 0.734 | 0.315 | 0.478 | 0.353 | 0.506 |
| SGA(gpt-4o-mini) | 0.747 | 0.667 | 0.875 | 0.577 | 0.421 | 0.603 |
| SGA(Qwen2.5-7B) | 0.535 | 0.686 | 0.611 | 0.465 | 0.286 | 0.472 |
| **Symbolic-R1 (Ours)** | **0.973** | 0.977 | **0.997** | 0.972 | **0.955** | **0.969** |

As presented in Tab. 8, our proposed method, Symbolic-R1, demonstrates state-of-the-art performance and robust generalization across five well-established benchmarks. It achieves the highest overall average score of 0.969, outperforming all competing methods. Specifically, Symbolic-R1 secures the top rank on three of the five benchmarks (Nguyen, R, and SRBench) and the second-best rank on the remaining two (Constant and Keijzer), consistently placing it at the forefront of performance. This strong result, which surpasses various baselines including recent LLM-based methods and is highly competitive with the powerful PySR, underscores the superior effectiveness and generalization capability of our approach.

### D.4 RESULTS VISUALIZATION

To provide a qualitative assessment of our method's performance, we present a case study in Table 9. The objective is to recover the ground-truth equation $C * x_1 + C * arctan(C * x_0 + C) + C$. As shown, our proposed method, Symbolic-R1, is the only one capable of precisely identifying the exact symbolic form of the ground-truth equation.

Furthermore, as shown in Table 10, we visualize more results on the SymbArena test set.

Table 9: A Visual Comparison with Baseline Methods.

| Method | Pred Equation |
|---|---|
| **Symbolic-R1(Ours)** | $C * x_1 + C * arctan(C * x_0 + C) + C$ |
| gplearn | $x_0 * *0.5 * (x_0 + C) * log(((x_0 * *0.5 * (x_0 * (C + x_1 * *2 * C/x_0) + C) * log(((x_1 + C) * log(((x_1 + C) * (x_1 + log(x_1 + C))/x_0) * *0.5)/x_0) * *0.5) + C) * log(x_0/((x_1 + C) * (x_1 + log(x_1 + C))/x_0) * *0.5)/x_0) * *0.5)$ |
| AFP | $C * x_0 * *3 + C * x_0$ |
| AFP-FE | $C * x_0 * *3 + C * x_0$ |
| GP-GOMEA | $x0 * (C + x0) * log(C/x0)$ |
| uDSR | $x_0 * (x_0 + exp(C))$ |
| PySR | $C * sin(sin(C * sin(x0))) + exp(C * x0)$ |
| LLM-SR(gpt-3.5-turbo) | $C * x_1 * *2 + C * x_2 * *2 + C * sin(x_1) + c * cos(x_2) + C$ |
| LLM-SR(gpt-4o-mini) | $C * x_1 * x_2 + C * x_1 + C * x_2 + C$ |
| LLM-SR(Qwen2.5-7B) | $C * x_1 * x_2 + C * x_1 + C * x_2 + C$ |
| SGA(gpt-3.5-turbo) | $C * x_0 * *2 + C * x_0 * x_1 + C * x_0 + C * x_1 * *2 + C * x_1 + C$ |
| SGA(gpt-4o-mini) | $C * x_0 + C * x_1 + C$ |
| SGA(Qwen2.5-7B) | $C * x_0 * *2 + C * x_0 + C * x_1$ |

Table 10: Example of results of Symbolic-R1

| GT Equation | Pred Equation |
|---|---|
| $C * x_0 * *2 + C$ | $C * x_0 * *2 + C$ |
| $C + C * x_0/x_1 + C * x_2/x_1$ | $C * x_0/x_1 + C + C * x_2 * *2/x_1 + C * x_2/x_1$ |
| $C * x_0 * arctan(C * x_0 + C) + C * x_0 + C$ | $C * x_0 * *2 + C * x_0 + C + (C * x_0 + C) * arctan(C * x_0 + C)$ |
| $C + (C * x_0 + C * sin(C * x_0 + C))/x_0$ | $C + C * sin(C * x_0 + C)/x_0$ |
| $C * x_0 + C * (C * x_0 + C) * *2 + C$ | $C * x_0 * *3 + C * x_0 + C * (C * x_0 + C) * *2 + C$ |
| $C + (C * x_0 * *2 + C * x_0)/x_0$ | $C * x_0 + C$ |

