# OpenReview forum: "Finetuning Large Language Model as an Effective Symbolic Regressor"
_ICLR.cc/2026/Conference — ICLR 2026 Conference Withdrawn Submission_

### Official Review · Reviewer_zGeA · 2025-10-30

**Soundness:** 2
**Presentation:** 2
**Contribution:** 1
**Rating:** 2
**Confidence:** 4

**Summary:**

This paper proposes fine-tuning a general-purpose LLM (Qwen2.5-7B) on a synthetic corpus of ~148k symbolic equations (“SymbArena”) to improve symbolic regression (SR). The training uses supervised instruction tuning plus a rule-based RL stage (“Form-GRPO”). Additionally, the paper proposes a new inference-time strategy, "Hypothesis–Experiment–Revision," to further improve the results. Their main experiments on a held-out test set of SymbArena show that the fine-tuned LLM achieves the best performance compared to other Symbolic Regression methods.

**Strengths:**

- Exploring LLMs for Symbolic regression is an interesting topic, as it in theory LLMs allows for combining textual and numerical information.
- Their ablation study on the different training stages and inference strategies is extensive.
- Their method is straightforward and easy to understand.

**Weaknesses:**

- The main weakness is the reliance on the Symbolic-R1 test set for the main results. As this set shares the same distribution as the training data, improved performance is expected. This leaves unanswered the core question of whether performance generalizes to other distributions.
- While the paper presents the new dataset as a contribution, the data creation pipeline itself is not novel. Consequently, the paper's primary contribution is the multiple step strategy of finetuning the LLM on this symbolic dataset which is rather weak.
- The absence of experiments leveraging both text and numerical data is a missed opportunity. The true power of an LLM would be demonstrated by incorporating textual information or priors, rather than merely competing with GP on numerical tasks.
- The paper lacks an analysis of catastrophic forgetting in the fine-tuned LLM. It is unclear if the model retains its ability to perform other tasks. If this is not the case, what is the motivation of using an LLM for this task?

**Questions:**

The appendix (Table 8) indicates the method was also tested on other benchmarks. While the results look interesting, the specific metric is not specified, as far as I can see. Can you report R>0.99, R>0.999, R>0.999 on this? Furtheremore, why wasn't this information placed in the main body?

Could you perform additional studies with datasets that are completely out-of-distribution? In general, I would be curious to understand how much the LLM is truly generalizing versus how much it is memorizing. Are the equations generated by the LLM, for the most part, present in the training set?

---

### Official Review · Reviewer_6dC2 · 2025-10-31

**Soundness:** 2
**Presentation:** 2
**Contribution:** 2
**Rating:** 2
**Confidence:** 5

**Summary:**

This paper proposes Symbolic-R1, an LLM-based symbolic regression method trained on SymbArena, a new benchmark with 148K synthetic equations. The approach uses instruction tuning followed by reinforcement learning (Form-GRPO) with structure-aware rewards, and employs iterative refinement (HER) during inference. The paper introduces a form-level consistency metric alongside numerical accuracy metrics. Results show improvements over baselines including PySR on the proposed benchmark.

**Strengths:**

* The core idea of fine-tuning LLMs specifically for symbolic regression is interesting and relatively underexplored compared to inference-time scaling approaches.
* The GRPO training scheme with multiple reward types is a clever design that tries to balance structural correctness with numerical accuracy.
* The reality-verification step for test equations (using LLM to check similarity to known physics) to ensure practical relevance of the benchmark seems novel and promising.

**Weaknesses:**

**Problem formulation.** The paper focuses on traditional SR without relying on domain knowledge (finding equations from data only) but the positioning is confusing. LLM-SR and SGA are designed for context-rich problems and seem tested outside their scope here. If we consider the contribution as "fine-tuning" the model for SR, one naturally thinks of transformer-based methods like E2E which, while trained on much more data, struggle with symbolic recovery. So it is not clear if the advantage here is coming from LLM prior knowledge, the training method, or test set characteristics (see the next point)? The paper doesn't clearly establish what problem it's solving and against which baselines it should primarily be evaluated.

**Contamination concerns.** The test set is "reality-verified" to be similar to known physics equations, which seems to increase rather than decrease contamination risk; if equations are similar to known physics, the pre-trained LLM likely learned these patterns. Evaluation on famous benchmarks (Nguyen, Feynman equations in SRBench) that are almost certainly in LLM pre-training data raises questions about whether improvements come from genuine SR capability or memorization. Contamination should be studied not only with respect to SymbArena's train set but also the LLM's pre-training corpus.

**Evaluation setup.** Following the previous point, the benchmark datasets may not be suitable for proper evaluation of general SR methods (see questions section for some suggested datasets). For completeness, it would be good to also add relevant neural baselines like E2E and TPSR. Evaluating on complex, real-world problems without contamination concerns could better demonstrate performance.

**Minor presentation issues.** Several typos exist (e.g., "Defination" in Figure 1, "Hypothe- sis" in third contribution bullet point). Table 8 doesn't specify which metric is reported.

**Questions:**

1. The reality-verification ensures similarity to known physics, and benchmarks include famous equations likely in pre-training data. How was contamination assessed? Could results be provided for: a) SRBench black-box problems (real-world data with no known equations) compared to traditional (PySR) and transformer-based methods (E2E, TPSR)? b) LLM-SRBench datasets - Table 1 tags these as "LLM-based only" but they can be used for any SR method, so this characterization seems inaccurate.

2. How is the data generation different from Lample & Charton? Is the same code used? The claim that reality-checking makes it "fairer than directly using real data...avoiding the possibility that LLM has seen them during pre-training" while simultaneously ensuring similarity to existing equations needs clarification; this appears to increase contamination risk for LLMs.

3. LLM-SR and SGA are designed for context-rich problems but evaluated on context-free data, so they don't seem like appropriate comparisons. Wouldn't LaSR (Grayeli et al.), which integrates with PySR, be more suitable? Could comparison be provided, along with qualitative analysis on Feynman equations to understand whether the model recovers structure or leverages memorization?

4. Could you break down SRBench performance for Feynman equations, ODEs, and black-box problems separately, providing symbolic accuracy, numeric accuracy, complexity, and runtime for each category? Also, I would suggest to bring the results on standard benchmarks to the main body of the paper as they are more reliable than test set of synthetic SymbArena dataset, especially since the test data distribution could be very close to the training distribution as they are sampled from the same generator.

5. Table 9 shows identical or very similar LLM-SR and SGA outputs across different LLMs. Given different models and frameworks, one wouldn't expect identical outputs.

6. Given that various symbolic accuracy metrics have been studied (SymPy symbolic recovery, tree edit distance, LLM-as-judge), what specific advantage does the proposed form-level consistency metric provide? and why is it novel?

---

### Official Review · Reviewer_3Vai · 2025-11-02

**Soundness:** 2
**Presentation:** 3
**Contribution:** 2
**Rating:** 2
**Confidence:** 4

**Summary:**

The paper introduces Symbolic-R1, a fine-tuned LLM for symbolic regression, and SymbArena, a dataset of 148K synthetic equations. The method combines instruction tuning, reinforcement tuning (Form-GRPO), and a Hypothesis-Experiment-Revision (HER) inference loop.

**Strengths:**

+ The dataset scale is impressive and could become a useful SR resource.
+ The dual metric design (numeric + structural) addresses a genuine limitation of past SR benchmarks.
+ The overall pipeline is ambitious and well-motivated from an empirical standpoint.

**Weaknesses:**

- The core idea (IFT + GRPO + iterative refinement) is a straightforward combination of existing methods. I have serious concerns about the lack of algorithmic innovation.
- Synthetic-only evaluation. All experiments are on a self-generated dataset; no validation on SRBench, Nguyen, or real scientific equations. Claims of generalization are unsupported.
- The GPT-4o adjudicator is non-reproducible. Using a closed-source model to score results is scientifically unsound.
- The “reality enhancement” uses an LLM to curate data for another LLM, introducing methodological bias.
- Traditional SR baselines are under-tuned. The authors should perform compute-matched comparisons and include modern neural SR baselines.
- R² for the same model differs across tables (0.201 vs 0.313). The authors must clarify this discrepancy.
- Key components (reward weighting, HER pseudocode, coefficient optimization) are missing. The paper is not reproducible.
- The contribution of each reward term and HER iteration depth is unclear.
- Numerous grammatical errors and confusing phrasing make sections hard to follow. Figures overstate the improvements.
- Overall, I appreciate the dataset effort, but this paper lacks novelty, rigor, and transparency. The evaluation setup is biased and confined to synthetic data, making the claimed generalization unreliable.

**Questions:**

1. How distinct are train/test skeletons? Provide a quantitative measure.
2. Why rely on GPT-4o for scoring instead of a reproducible rule metric?
3. Can Symbolic-R1 handle equations outside the fixed operator library?
4. What is the actual compute cost of training and inference?
5. Which component (IFT, RFT, or HER) contributes most to the gain?

---

### Official Review · Reviewer_RtLW · 2025-11-02

**Soundness:** 1
**Presentation:** 3
**Contribution:** 2
**Rating:** 2
**Confidence:** 4

**Summary:**

This paper is focused on the problem of symbolic regression using LLMs by introducing a large-scale synthetic benchmark called SymbArena and a fine-tuned model named Symbolic-R1. The authors argue that pre-trained LLMs underperform on SR due to the mismatch between their approximate reasoning capabilities and the precision demands of equation discovery task. To address this, they propose a two-stage fine-tuning pipeline (instruction tuning + reinforcement tuning via Form-GRPO) and an inference-time iterative strategy (Hypothesis–Experiment–Revision, or HER). Their experiments show improvements over both traditional non-LLM SR algorithms and prior LLM-based methods.

**Strengths:**

- The paper is generally well-written and easy to follow.
- Figures and visual explanations (especially Figure 3) clearly present the overall pipeline and make the methodology understandable.
- The motivation of enhancing LLM backbones for symbolic regression through synthetic fine-tuning is well-motivated and interesting.

**Weaknesses:**

- What test data is used for the results in Tables 2 and 3? If the evaluations are conducted only on SymbArena, even with a held-out split, testing on data from the same generation distribution does not convincingly demonstrate generalization or fair comparison to other baselines. I would like to see evaluation of the fine-tuned model on LLM-SRBench [1], which provides a more diverse and domain-general benchmark for LLM-based equation discovery.
- Since LLM-SRBench also reports symbolic accuracy (SA), I would like to see how the finetuned model performs on this benchmark with respect to that metric, and how it compares to both existing baselines and the authors’ own proposed metrics. This would help assess the effectiveness of the proposed contributions in a more standardized setting
- In Table 3, why does the +IFT model outperform the +IFT+RFT variant on some metrics (e.g., the LLM-as-judge metric in the first column)? If the proposed form-consistency metric is indeed well-aligned with the task objectives, one would intuitively expect improvements after RL fine-tuning. Could the authors clarify the reason behind this result?
- In Table 3, what exactly does the HER (hypothesis–experiment–revision) framework represent without fine-tuning? If it only involves inference-time iterations, how does it differ from prior methods such as LLM-SR or SGA? It appears to be a simple iterative refinement process without an explicit external memory or buffer to guide exploration. Given that, I’m quite surprised that this inference-only variant outperforms other evolutionary search approaches that explicitly maintain such buffers to avoid local optima (eg LLM-SR). Could the authors elaborate on the underlying reason for this result? This point is not clearly explained in the current version of the paper.


**Minor Comments:**
- In Sec 2.2 (L274), the authors describe six features used for computing the form-level consistency metric as an alternative to the binary equivalence metric. However, it is unclear what exactly these "structural patterns" refer to and how they are defined. If they are based on combinations of symbolic terms, how are these terms selected, and how many such combinations are considered? Providing concrete examples of these structural patterns and explaining how they are extracted from the generated hypotheses is needed for clarity.
- In Table 2 and Sec 4/4.2, there is no description of the non-LLM baselines (other than PySR). It would be helpful to include a brief summary and appropriate references for these baselines within the main paper or refer to appendix for more details.


[1] LLM-SRBench: A New Benchmark for Scientific Equation Discovery with Large Language Models, ICML 2025

**Questions:**

Included in the weaknesses

---

### Note · Authors · 2025-11-27

I have read and agree with the venue's withdrawal policy on behalf of myself and my co-authors.